# RNA Pol IV induces antagonistic parent-of-origin effects on *Arabidopsis* endosperm

**Prasad R. V. Satyaki**[1], **Mary Gehring**[1,2]*

**1** Whitehead Institute for Biomedical Research, Cambridge, Massachusetts, United States of America,
**2** Department of Biology, Massachusetts Institute of Technology, Cambridge, Massachusetts, United States of America

* mgehring@wi.mit.edu

**Data Availability Statement:** Most relevant data is within the paper and its Supporting Information files. High-throughput sequencing data is deposited in NCBI GEO accession GSE197717.

## Abstract

Gene expression in endosperm—a seed tissue that mediates transfer of maternal resources to offspring—is under complex epigenetic control. We show here that plant-specific RNA polymerase IV (Pol IV) mediates parental control of endosperm gene expression. Pol IV is required for the production of small interfering RNAs that typically direct DNA methylation. We compared small RNAs (sRNAs), DNA methylation, and mRNAs in *Arabidopsis thaliana* endosperm from heterozygotes produced by reciprocally crossing wild-type (WT) plants to Pol IV mutants. We find that maternally and paternally acting Pol IV induce distinct effects on endosperm. Loss of maternal or paternal Pol IV impacts sRNAs and DNA methylation at different genomic sites. Strikingly, maternally and paternally acting Pol IV have antagonistic impacts on gene expression at some loci, divergently promoting or repressing endosperm gene expression. Antagonistic parent-of-origin effects have only rarely been described and are consistent with a gene regulatory system evolving under parental conflict.

## Introduction

Parents influence zygotic development in viviparous plant and animal species. In flowering plants, parent-of-origin effects on offspring development are observed in an embryo-surrounding seed tissue called the endosperm [1]. Endosperm does not contribute genetic material to the next generation but mediates maternal nutrient transfer to the embryo, coordinates growth between the embryo and maternal tissues, sets seed dormancy and regulates germination, and acts as a nutrient store to support seedling growth [2]. Endosperm is typically triploid and develops from the fertilization of a diploid female gamete, called the central cell, by 1 of 2 haploid sperm cells that are released by pollen. Violations of the balanced ratio of 2 maternal to 1 paternal genomes disrupts normal endosperm development in a parent-of-origin–dependent manner [3–7]. In some *Arabidopsis thaliana* accessions, crosses between tetraploid mothers and diploid fathers exhibit reduced endosperm proliferation and smaller mature seeds, while reciprocal crosses where the fathers are tetraploid (paternal excess crosses) exhibit prolonged endosperm proliferation and larger or aborted seeds. These parent-of-origin effects on endosperm development have been interpreted under the aegis of the parental conflict or

**Funding:** This research was supported by National Science Foundation (NSF) grants 2101337 and 1453459 to M.G. The funders had no role in study design, data collection and analysis, decision to publish, or preparation of the manuscript.

**Competing interests:** The authors have declared that no competing interests exist.

**Abbreviations:** ISR, imprinted sRNA region; mRNA-seq, mRNA sequencing; Pol IV, polymerase IV; PRC2, polycomb repressive complex2; QTL, quantitative trait locus; RdDM, RNA-directed DNA methylation; RT-PCR, reverse transcription PCR; sRNA, small RNA; TE, transposable element; WT, wild-type.

kinship model [8,9]. According to this model, when a mother mates with more than 1 father, the inclusive fitness of the mother may be optimized if her resources are equally distributed among her progeny, to which she is equally related. The inclusive fitness of the father is optimal when his progeny are able to acquire more finite maternal resources than other half-siblings. Such conflicts are postulated to lead to arms races whose impacts may be observed in the molecular machinery mediating parental control. However, our understanding of the impact of conflict on endosperm biology is limited by our incomplete understanding of molecular and genetic mechanisms guiding parental control of endosperm development.

Recent data indicate that small RNAs (sRNAs) and mutations in RNA polymerase IV (Pol IV) have effects on reproduction, endosperm, and seed development in multiple species [10–16]. RNA Pol IV functions as part of the RNA-directed DNA methylation (RdDM) pathway, in which it produces relatively short, noncoding transcripts that are converted into double-stranded RNA by RDR2 [17–19]. These double-stranded RNAs are cleaved into 24-nt sRNAs by DCL3, and single strands are loaded into ARGONAUTE proteins that help target the de novo DNA methyltransferase DRM2, which acts in conjunction with RNA Pol V and several other proteins, to methylate DNA [20]. *NRPD1*, which encodes the largest subunit of RNA Pol IV, has roles in endosperm gene dosage control. Endosperm gene expression typically reflects the ratio of 2 maternally and 1 paternally inherited genomes, such that for the majority of genes approximately two-thirds of transcripts are derived from maternal alleles [21,22]. A survey of allele-specific gene expression in *nrpd1* mutant endosperm found that Pol IV is required to maintain the 2:1 maternal to paternal transcript ratio in the endosperm and that loss of Pol IV leads to the misregulation of several hundred genes [14]. Additionally, loss-of-function mutations in *NRPD1* or other members of the RdDM pathway can repress seed abortion in crosses of diploid mothers and tetraploid fathers [10,14,15]. In *Brassica rapa*, loss of *NRPD1*, *RDR2*, or *NRPE1* results in high rates of seed abortion due to maternal sporophytic effects [13]. Loss of Pol IV in both *B. rapa* and *A. thaliana* also results in smaller seed sizes [13], and RNA Pol IV is essential for postmeiotic pollen development in *Capsella rubella* [12].

Molecular data point to the intriguing possibility that mutations in RNA Pol IV have parent-of-origin effects on endosperm. A comparison of sRNAs in wild-type (WT) whole seeds (which includes maternal seed coat, endosperm, and embryo) with *NRPD1*+/− endosperm from crosses where the mutation in *NRPD1* was either maternally or paternally inherited suggested that loss of maternal *NRPD1* affected more sRNA loci than the loss of paternal *NRPD1* [11]. Although the comparison of sRNAs from WT whole seeds to mutant endosperm in this study makes definitive conclusions difficult to draw, it raises the potential question of if and how the loss of *NRPD1* has parent-of-origin effects on sRNA production.

To examine the impacts of parental Pol IV activity on endosperm in more detail, we examined sRNA and mRNA transcriptomes in WT endosperm, *nrpd1* homozygous mutant endosperm, and *nrpd1* heterozygous endosperm where the mutant allele was inherited from a homozygous mutant mother or father. We also examined endosperm methylomes in WT and in the reciprocal heterozygotes. Analyses of these data demonstrate that maternal and paternal *NRPD1* have distinct parental effects on endosperm, some of which are antagonistic.

## Results

### Maternal Pol IV inhibits, whereas paternal Pol IV promotes, interploidy seed abortion

We tested if the molecular data supporting distinct functions for Pol IV in the mother and the father [11] could be supported by genetic analyses. We reanalyzed previously published data [14] to specifically test the effects of the loss of maternal Pol IV versus paternal Pol IV in the

context of interploidy, paternal excess crosses (diploid mother pollinated by tetraploid father). Inheritance of a mutant *nrpd1* allele from diploid mothers resulted in 4% normal seed in a cross to tetraploid fathers, which was significantly different than 7.1% normal seed observed when WT diploid mothers are crossed to WT tetraploid fathers. Crosses between WT diploid mothers and tetraploid *nrpd1* fathers resulted in 64.8% normal seed. Paternal rescue by *nrpd1* was diminished when the diploid mother was also mutant, resulting in 37.5% normal seed. Thus, we conclude that maternal *NRPD1* promotes interploidy seed viability and paternal *NRPD1* represses seed viability (S1 Fig). This is consistent with observations that paternal excess seed viability was promoted by the maternal activity of *DCL3* and repressed by the paternal activity of *DCL3* (*DCL3* functions downstream of *NRPD1*) [10]. Interploidy crosses are a sensitive genetic assay to detect endosperm phenotypic effects. However, paternal excess endosperm displays widespread transcriptomic changes [10], which make it a poor system to understand the specific role of RNA Pol IV in endosperm development. Therefore, for all subsequent experiments, we examined endosperm molecular phenotypes in the context of balanced crosses (diploid × diploid) where either one or both parents were homozygous mutant for the *nrpd1a-4* allele.

## Loss of maternal or paternal Pol IV activity impacts sRNAs at distinct sites

To determine the role of Pol IV in sRNA production in the endosperm, we first identified Pol IV–dependent endosperm sRNAs. Previously, we showed that 24-nt sRNAs were the predominant sRNA species in endosperm and exhibited a broader distribution over genes and transposable elements (TEs) than in other tissues [14]. We profiled sRNA populations in 3 replicates of endosperm derived from crosses of L*er nrpd1* females pollinated by Col-0 *nrpd1* males (7 days after pollination) and compared them with our previously published sRNA libraries from L*er* × Col-0 WT $F_1$ endosperm (female parent in cross written first) [14] (S1 Table).

We identified 21,131 sRNA peaks in WT endosperm using ShortStack [23]. A total of 76.9% of these were predominantly populated by 24-nt sRNAs, with another 1.1%, 0.2%, and 2.2% of peaks predominated by 23-, 22-, or 21-nt sRNAs, respectively. An additional 19.7% of peaks were either predominated by a noncanonical sRNA size or had no predominant size class (S2A Fig). The majority of sRNAs were genetically dependent on *NRPD1*, with 99% of 24-nt sRNA peaks, 94.87% of 22-nt sRNA peaks, and 70.1% of 21-nt sRNA peaks absent in *nrpd1*−/− endosperm (S2B Fig). To enable downstream comparisons to expression, we binned sRNAs by size (21 to 24 nt) and calculated read counts overlapping TEs and genes encoding proteins, miRNAs, and other ncRNAs. We used DESeq2 [24] to separately identify genes and TEs with significant differences in Pol IV–dependent sRNA populations. Consistent with the peak-based analysis, loss of RNA Pol IV abolished 21- to 24-nt sRNAs at most TEs and genes, while most miRNAs were not impacted (S2 and S3 Tables, S3 Fig). Moreover, 21- to 23-nt sRNAs were often lost at the same loci as 24-nt sRNAs (S2C Fig), suggesting that sRNAs of differing sizes arose from the same Pol IV transcript in the WT but were likely processed into RNAs shorter than 24 nt by different downstream DICERs or by the exosome components *Atrimmer1* and *2* [25,26]. Pol IV–dependent 21- to 23-nt sRNAs have been identified in other tissues, indicating that this finding is not specific to endosperm [12,27,28].

After identifying Pol IV–dependent sRNAs, we asked whether loss of one parent's Pol IV influenced the abundance of Pol IV–dependent sRNAs in *nrpd1* heterozygous endosperm. We sequenced sRNAs from 2 replicates of L*er* female × Col-0 *nrpd1*−/− male (referred to as pat *nrpd1+/−)* $F_1$ endosperm and 3 replicates of L*er nrpd1*−/− female × Col-0 male (referred to as mat *nrpd1+/−)* $F_1$ endosperm. Because the endosperm is triploid, in these comparisons, there

are 3 (WT), 2 (pat *nrpd1*+/−), 1 (mat *nrpd1*+/−), and 0 (*nrpd1*−/−) functional *NRPD1* alleles in the endosperm. However, expression of *NRPD1* is paternally biased in WT L*er* × Col endosperm [21]. Consistent with paternal allele bias, mRNA sequencing (mRNA-seq) data show that *NRPD1* is expressed at 42% of WT levels in pat *nrpd1*+/− and at 91% of WT levels in mat *nrpd1*+/− endosperm (S10 Table).

We found that the presence of functional *NRPD1* inherited from either parent is sufficient for the biogenesis of nearly WT levels of 21- to 24-nt sRNAs in endosperm (Fig 1A, S3 Fig). However, although the overall sRNA population in the heterozygotes was similar to the WT (Fig 1A), loss of maternal and paternal *NRPD1* had distinct impacts on sRNA at individual loci (Fig 1B–1F, S3 Fig, S4–S7 Tables). We identified genes and TE insertions that displayed at least a 2-fold change in the abundance of sRNAs in mat or pat *nrpd1*+/− compared to the WT (Fig 1B–1F, S3 Fig, S4–S7 Tables). Loss of paternal *NRPD1* caused relatively small fold change reductions in 21- to 24-nt Pol IV sRNAs at a handful of loci, while loss of maternal *NRPD1* had slightly greater yet limited impact (Fig 1B–1F, S3 Fig, S4–S7 Tables). For genic loci with *NRPD1*-dependent 24-nt sRNAs, 2% (327 genes) had significantly lower sRNA abundance in mat *nrpd1*+/− compared to WT; in contrast, 0.3% (60 genes) had significantly lower sRNAs in pat *nrpd1*+/− (Fig 1B). For TE loci with *NRPD1*-dependent 24-nt sRNAs, 2.8% (545 TE insertions) and 1.35% (261 TE insertions) exhibited significantly lower sRNA abundance in mat and pat nrpd1+/−, respectively (Fig 1B). Few of the loci with reduced sRNAs were shared between the reciprocal heterozygote—of 327 24 nt sRNA-expressing genic loci that were reduced by more than 2-fold in mat *nrpd1*+/−, only 22 were also reduced by 2-fold in pat *nrpd1*+/− (Fig 1F). Moreover, there was no quantitative or correlative relationship between loci affected in mat *nrpd1*+/− and pat *nprd1*+/− (Fig 1F). Thus, the vast majority of sRNA-producing loci in endosperm only require at least 1 functional copy of *NRPD1* after fertilization.

## Evaluating memory of parental Pol IV activity and endosperm sRNA production

The absence of dramatic differences in sRNAs in heterozygotes could indicate that the alleles inherited from both the WT and the *nrpd1*−/− parent produce a WT level of sRNAs after fertilization. This result would be expected for a recessive mutation without parental effects. However, it is known that Pol IV activity at some loci requires prior Pol IV activity [29]. Under such a scenario, Pol IV activity in the parents before fertilization might be necessary for sRNA production from that parent's allele in the endosperm after fertilization. Thus, the observed lack of differences in sRNA production at most loci in heterozygous *nrpd1* endosperm (Fig 1) might be explained by an up-regulation of sRNA production from the alleles inherited from the WT parent (i.e., sRNAs are up-regulated from paternal alleles in mat *nrpd1*+/−, and maternal allele sRNAs are up-regulated in pat *nrpd1*+/− endosperm). To distinguish between these possibilities, we used the SNPs between Col-0 and L*er* to identify the allelic origins of sRNAs in WT and heterozygous endosperm. We first confirmed prior observations that Pol IV sRNAs are biallelically expressed at most loci in endosperm and predominantly expressed from one parental allele, or imprinted, at several hundred others [14]. By examining sRNAs at genes and TEs, we found that both biallelically expressed 21- and 24-nt sRNA loci (defined as between 20% and 80% of sRNAs from maternally inherited alleles) and those predominantly expressed from one parental allele (>80% or <20% maternal) were Pol IV-dependent (i.e., their accumulation was significantly reduced in *nrpd1*−/− endosperm) (S4 Fig).

To test if sRNA production from alleles inherited from WT parents compensated for alleles inherited from an *nrpd1*−/− parent, we first assessed several thousand loci that were not significantly misregulated in *nrpd1*+/− endosperm. Overall, there were similar contributions from

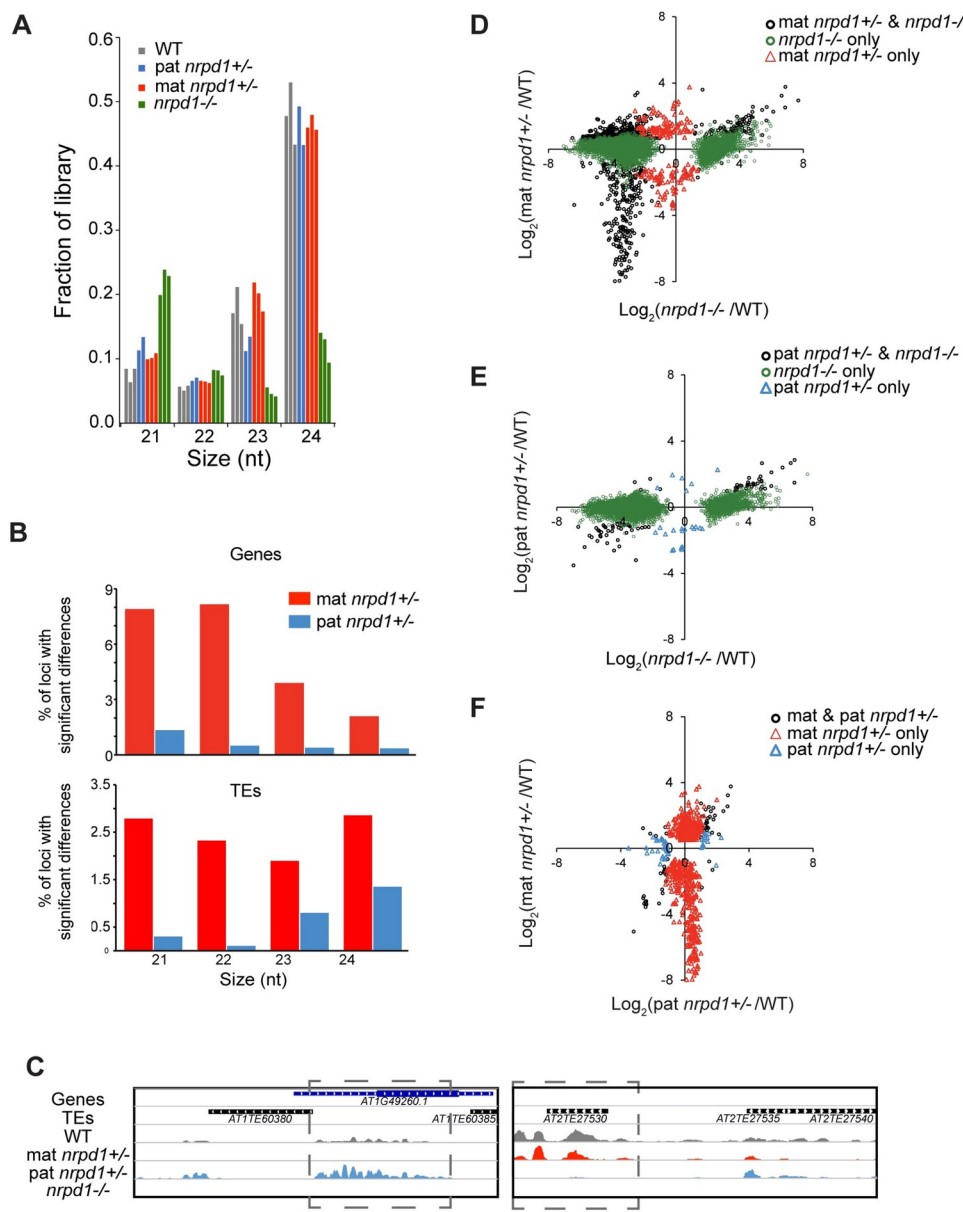

**Fig 1. Impact of loss of maternal, paternal, or both copies of *NRPD1* on endosperm sRNAs. (A)** Loss of maternal or paternal *NRPD1* does not substantially alter the endosperm sRNA pool. Fraction of aligned sRNA reads in each size class in the indicated genotypes. **(B)** Examination of 21- to 24-nt sRNAs over genes or TEs shows that inheriting a mutant maternal *nrpd1* allele has a greater impact than inheriting a mutant paternal *nrpd1* allele. Percent of loci showing at least a 2-fold reduction in sRNA abundance and *padj* < 0.05 according to DESeq2 are indicated in red (mat *nrpd1*+/−) or blue (pat *nrpd1*+/−). Genes and TEs included in this tally have a normalized WT read count of 5 or higher. **(C)** Snapshots of loci with Pol IV–dependent 24-nt sRNAs that show a specific loss of sRNAs in mat (left) or pat (right) *nrpd1*+/− endosperm. **(D–F)** Comparisons of genic 24-nt sRNAs upon loss of maternal, paternal, or both copies of *NRPD1*. Fold change as calculated by DESeq2. Only significant changes (*padj* < 0.05) are plotted. Underlying data for Fig 1 A, B, and D–F can be found in S1 Data. Pol IV, polymerase IV; sRNA, small RNA; TE, transposable element; WT, wild-type.

maternal and paternal alleles in mat and pat *nrpd1* heterozygotes compared to WT endosperm (Fig 2A). This suggests that by 7 DAP (days after pollination), at most loci in the endosperm, sRNAs are produced from both maternal and paternal alleles regardless of whether the alleles

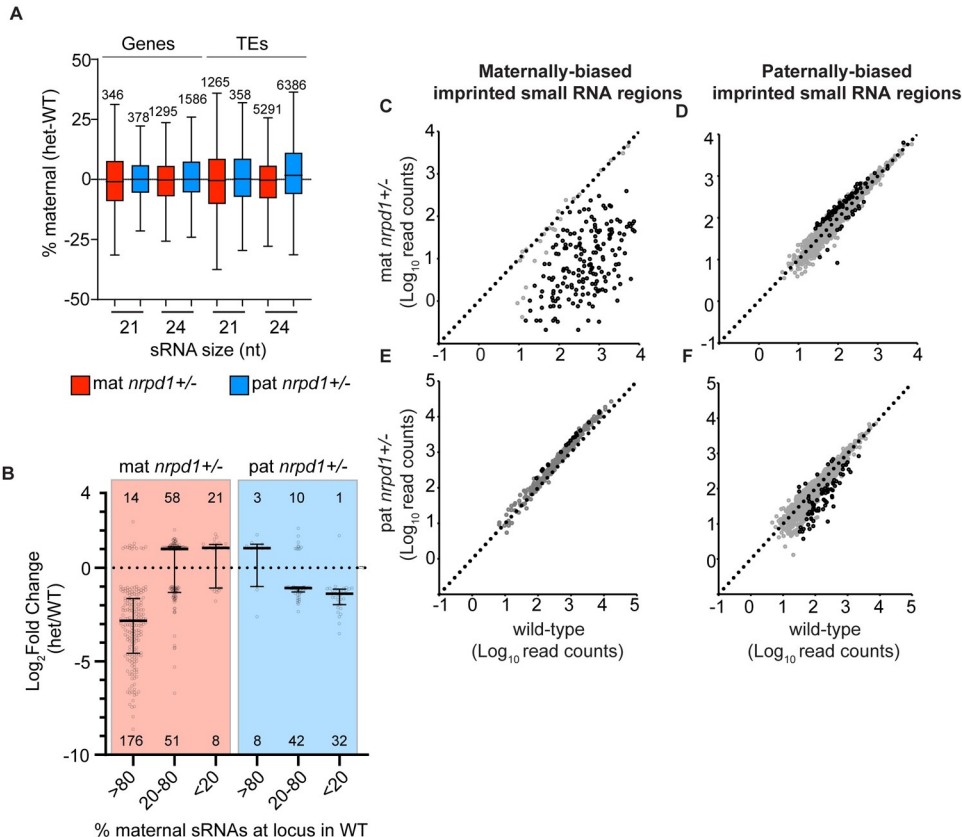

**Fig 2. Effects of loss of maternal or paternal Pol IV on the allelic origin of sRNAs. (A)** Tukey plot shows no difference in allelic origin of genic and TE sRNAs between *nrpd1* heterozygotes and WT. Loci plotted here showed similar sRNA abundances in WT and heterozygotes and have a sum of at least 10 allele-specific reads in 3 WT replicates and in heterozygotes. **(B)** Loci with reduced sRNAs in mat *nrpd1+/−* or pat *nrpd1+/−* exhibit maternally or paternally biased sRNAs in WT. Genes and TEs showing differential abundance of 24-nt sRNAs in *nrpd1* heterozygotes were grouped into bins by the % of sRNAs produced from the maternal alleles of that locus in WT. Fold change was calculated by DESeq2. Tukey plot represents fold change in each group; circles show fold change at individual loci. Numbers below and above plot are total number of loci having significantly lower and higher abundance of 24-nt sRNAs in *nrpd1+/−* relative to the WT. **(C–F)** Loss of maternal *NRPD1* leads to a reduction in the abundance of sRNAs from maternally biased ISRs (C) and gain of sRNAs from a subset of paternally biased ISRs (D). Loss of paternal *NRPD1* has a negligible impact on maternally biased ISRs (E) and a relatively minor impact on paternally biased ISRs (F). The Col-L*er* ISRs used here were defined by Erdmann and colleagues (2017). To identify regions with changes in sRNA abundance, read counts were calculated over sliding windows of 300 bp with 200-bp overlap. Windows with differential abundance were identified using DESeq2. Windows overlapping an ISR were identified using bedtools intersect. Overlapping windows were merged using bedtools merge and the median read count for each set of merged windows was plotted. Windows with and without significant differences in abundance are represented by black and gray circles. Data represented in this figure can be found in S3 Data. ISR, imprinted sRNA region; Pol IV, polymerase IV; sRNA, small RNA; TE, transposable element; WT, wild-type.

were inherited from a WT parent or an *nrpd1−/−* parent. However, we found that imprinted sRNA regions (ISRs; 113.1-kb maternally imprinted and 1,215.6-kb paternally imprinted regions overlapping both genic and TE loci) [14] were impacted by loss of parental Pol IV (Fig 2C–2F). ISR loci have been filtered to remove regions that are also enriched for seed coat sRNAs [14] and thus preclude analytical artifacts that may arise due to maternal tissue contamination or due to any potential sRNA movement. A total of 179 of 206 ISRs where expression was maternally biased in the WT showed reduced 24-nt sRNAs in mat *nrpd1+/−* (Fig 2C). On the other hand, only a small subset (74 of 2,405 ISRs) of paternally biased ISRs produced fewer sRNAs in pat *nrpd1+/−* (Fig 2F) and slightly more sRNAs in mat *nrpd1+/−* (Fig 2D). We also

note that maternally biased regions in WT showed slightly elevated production of sRNAs in pat *nrpd1+/−* endosperm (Fig 2E). Examination of the allelic origins of sRNAs at genes and TEs are also consistent with the ISR analysis. We found that sRNA loci showing dramatic reductions in abundance in mat *nrpd1+/−* tended to be maternally biased in WT endosperm (>80% of sRNAs from the maternally inherited alleles) (Fig 2B, leftmost column). Similarly, in pat *nrpd1+/−*, paternally biased sRNA (<20% sRNAs from the maternally inherited alleles) loci were more impacted (Fig 2B, rightmost column).

In summary, these results indicate that most maternally and some paternally biased imprinted sRNA loci in endosperm are dependent on Pol IV activity in the parents and are not established de novo postfertilization. Notably, these sites of Pol IV action are by definition distinct between maternal and paternal parents.

## Maternal and paternal RNA Pol IV have antagonistic impacts on gene expression

We previously identified several hundred genes misexpressed in *nrpd1−/−* endosperm [14]. To test for maternal or paternal effects on endosperm gene expression, we performed mRNA-seq in 3 replicates each of mat *nrpd1+/−* and pat *nrpd1+/−*, along with appropriate WT controls and homozygous mutant *nrpd1* endosperm (S1 Table). Evaluation of these datasets using a tissue enrichment test showed no indication of contamination with maternal seed coat tissue (S5 Fig). Differential expression analyses identified 1,791 genes whose transcripts were more abundant and 1,455 that were less abundant in *nrpd1−/−* compared to WT endosperm (Fig 3, S10 Table). Almost 50% of these genes (1,599) were similarly misregulated in mat *nrpd1+/−* (Fig 3A and 3B), along with 2,998 additional genes. In contrast, very few genes (90) changed in expression in pat *nrpd1+/−* compared to the WT (Fig 3A and 3B). In addition to the difference in the size of the effect, loss of maternal or paternal Pol IV altered the expression of different classes of genes. Panther overrepresentation tests [30] indicated that in mat *nrpd1+/−*, down-regulated genes were enriched for functions in the cell cycle, whereas up-regulated genes were enriched for functions in photosynthesis, stress response, and abscisic acid signaling (S11 Table). In pat *nrpd1+/−*, up-regulated genes were enriched for functions in heat stress response, while down-regulated genes were enriched for functions in responses to fungi (S11 Table). The expression of imprinted genes is known to be regulated epigenetically in endosperm. In mat *nrpd1+/−* endosperm, imprinted genes were more likely to be misregulated than expected by chance (hypergeometric test $p < 10^{-15}$)—15 out of 43 paternally expressed and 45 out of 128 maternally expressed imprinted genes were misregulated in mat *nrpd1+/−*, whereas 2 maternally expressed imprinted genes but no paternally expressed imprinted genes were misregulated in pat *nrpd1+/−* (S6 Fig, S10 Table).

Differential expression of a gene between WT and *nrpd1−/−* could represent (1) maternal and paternal effects arising from the loss of *NRPD1* in parents; (2) zygotic effects arising from epistatic interactions between mat *nrpd1−* and pat *nrpd1−*; (3) effects from the loss of all *NRPD1* in the endosperm; or (4) the sum of all 3 effects. As this study does not examine the effect of knocking down *NRPD1* specifically in the endosperm, we can only detect parental effects. Curiously, 2,988 genes misregulated in mat *nrpd1+/−* were not misregulated in *nrpd1−/−* endosperm (Fig 3A). We hypothesized that genic misregulation found exclusively in mat *nrpd1+/−* (but not *nrpd1−/−*) was caused by separate transcriptional effects of maternal and paternal *nrpd1* that were obscured in null mutants. To test this hypothesis, we compared gene expression between mat and pat *nrpd1+/−* (Fig 3C–3F). We found that 51/90 genes misregulated in pat *nrpd1+/−* endosperm were also misregulated in mat *nrpd1+/−* endosperm. However, 36 of these 51 genes changed expression in the opposite direction (hypergeometric

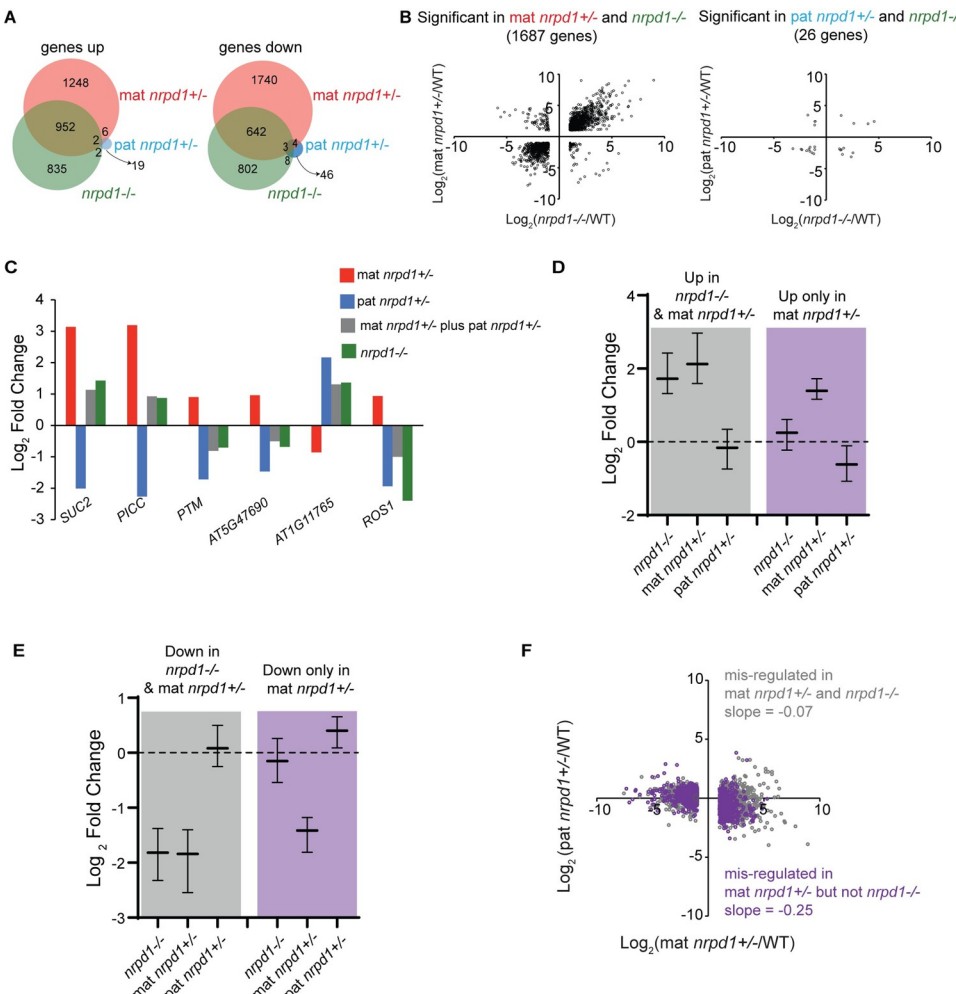

**Fig 3. Maternally and paternally acting Pol IV have antagonistic effects on endosperm gene expression. (A)** Venn diagrams showing overlap of genes with increased and decreased expression in comparison to WT endosperm for the indicated genotypes. **(B)** Scatter plots of genes that are significantly different (q ≤ 0.05, Log$_2$(Fold Change) ≥1 or ≤ −1) between WT and the indicated mutants. Fold change calculated using Cuffdiff. **(C)** Examples of genes that are antagonistically regulated by Pol IV. Gray bars represent mathematical sum of effects observed in mat and pat *nrpd1* +/−. **(D–F)** Inverse relationship between changes in gene expression in mat and pat *nrpd1*+/− relative to WT. Genes that are antagonistically influenced by maternal and paternal *NRPD1* are colored purple, while genes not antagonistically regulated are colored gray. In D, genes up-regulated at least 2-fold in both *nrpd1*−/− and mat *nrpd1* +/− do not exhibit misregulation in pat *nrpd1*+/−, while genes up-regulated only in mat *nrpd1*+/− but not *nrpd1*−/− have decreased expression in pat *nrpd1*+/−. In E, genes down-regulated in both *nrpd1*−/− and mat *nrpd1*+/− are not misregulated in pat *nrpd1*+/−, while genes down-regulated only in mat *nrpd1*+/− but not in *nrpd1*−/− are overall slightly increased in expression in pat *nrpd1*+/−. In F, genes that are significantly (2-fold, q ≤ 0.05) misregulated in both mat *nrpd1*+/− and *nrpd1*−/− show little to no inverse relationship between mat and pat *nrpd1*+/− (slope = −0.07), while genes that are only misregulated in mat *nrpd1*+/− but not *nrpd1*−/− are inversely affected in pat *nrpd1*+/− (slope = 0.25). Plots D and E show median and interquartile range for log$_2$(fold change) in mutant/WT. Fold change were calculated by Cuffdiff. Data represented in this figure can be found in S4 Data. Pol IV, polymerase IV; WT, wild type.

test for enrichment, $p < 10^{-10}$). For example, expression of the gene *SUC2* decreased about 4-fold in pat *nrpd1* +/− endosperm and increased about 8-fold in mat *nrpd1*+/− endosperm (Fig 3C). If *NRPD1* loss has no endospermic (zygotic) effect on the expression of these genes, then the misregulation observed in *nrpd1*−/− endosperm would be the sum of the parental effects. Indeed, the change in abundance of these genes in *nrpd1*−/− endosperm is close to that

predicted by an additive, antagonistic parental effect (compare gray and green bars in Fig 3C). *SUC2* transcript abundance in *nrpd1*−/− changes by 2.7-fold compared to the predicted 2.18-fold change, and other genes show similar effects (Fig 3C). While the expression of these particular genes showed large effects in both heterozygotes, most genes misregulated in mat *nrpd1*+/− did not show a significant change (defined as >2-fold difference in transcript abundance) in pat *nrpd1*+/− endosperm. We therefore hypothesized that misregulation of genes in mat *nrpd1*+/− but not *nrpd1*−/− endosperm was due to a small antagonistic effect arising from the loss of pat *NRPD1* in *nrpd1*−/−. To further test this hypothesis, we evaluated the expression of genes in pat *nrpd1*+/− endosperm that were either misregulated in both mat *nrpd1*+/− and *nrpd1*−/− or only in mat *nrpd1* +/− (Fig 3D–3F). Transcripts that significantly increased exclusively in mat *nrpd1*+/− had slightly decreased expression in pat *nrpd1*+/− endosperm (Fig 3D, purple). In contrast, genes that were significantly up-regulated in both mat *nrpd1*+/− and *nrpd1*−/− were not affected in pat *nrpd1*+/− (Fig 3D, gray). Similarly, genes that showed a significant reduction in abundance only in mat *nrpd1*+/− were slightly higher expressed in pat *nrpd1*+/− endosperm (Fig 3E, purple), while genes with reduced abundance in both mat *nrpd1* +/− and *nrpd1*−/− were not affected in pat *nrpd1*+/− (Fig 3E, gray). Finally, the antagonistic relationship could also be observed when directly comparing changes in mRNA abundance at individual genes, genome-wide, and upon loss of maternal and paternal Pol IV. Genes that were similarly misregulated in mat *nrpd1*+/− and *nrpd1*−/− showed a limited relationship (slope = −0.07). By contrast, genes that were misregulated in mat *nrpd1*+/− but not *nrpd1*−/− exhibited a clear inverse relationship (slope = −0.25) These results are consistent with an antagonistic parent-of-origin effect model for the impact of Pol IV on endosperm transcriptomes. Although the antagonistic effect at most genes is less than the commonly used 2-fold threshold difference for a significant change in gene expression, it is similar in magnitude to dosage compensation effects in other systems, such as that observed for the fourth chromosome in *Drosophila melanogaster* and for genes mediating genetic compensation in zebrafish [31,32].

## Evaluating possible mechanisms of Pol IV's impact on gene expression

How does Pol IV have parent-of-origin effects on gene expression in the endosperm after fertilization? Pol IV effects could be direct or indirect at the affected loci. One possibility is that Pol IV modulates gene expression via the proposed posttranscriptional gene silencing (mRNA cleavage) or translational inhibition by 21- to 22-nt Pol IV–dependent sRNAs [27,33]. Or, Pol IV–dependent RdDM over genic sequences or associated gene regulatory elements might repress transcription in the WT. Alternatively, Pol IV could impact many genes in *trans* by regulating the expression of chromatin proteins, like the known target *ROS1* (a DNA demethylase) (Fig 3C) [34], transcription factors [11,35], or by broadly influencing genome organization, which, in turn, affects gene expression [36,37]. To estimate the contribution of *cis*- and *trans*-effects and identify potential *cis*-regulatory targets of Pol IV that could drive widespread *trans*-effects, we analyzed the congruence of sRNAs, DNA methylation, mRNA cleavage patterns, and allele-specific changes driving gene expression changes in WT and mutant endosperm.

## Assessing potential mRNA cleavage by Pol IV–dependent sRNAs in endosperm

Pol IV–dependent genic sRNAs are proposed to regulate gene expression by cleaving mRNA [27], and we previously demonstrated that endosperm has greater accumulation of genic sRNAs than other tissues [14]. To test if such cleavage events contribute to endosperm Pol IV–

dependent transcript abundance, we first identified candidate genes that exhibited significantly increased ($\geq$2-fold) mRNA abundance and significantly decreased 21-, 22-, or 24-nt sRNA abundance ($\geq$2-fold) in *nrpd1−/−* endosperm (S7A–S7C Fig). This analysis suggested that at least 305 genes, or 16% of the genes that increase in expression in *nrpd1−/−* endosperm, were associated with Pol IV–dependent sRNAs. To directly assay if these genic sRNAs drive mRNA cleavage at levels sufficient to alter transcript abundance at specific loci, we mapped the 5′ ends of mRNA from WT and *nrpd1−/−* endosperm mRNA using NanoPARE sequencing [38]. NanoPARE maps both the 5′ ends of primary transcripts and those that result from mRNA cleavage. We confirmed that NanoPARE sequencing was working for us by identifying transcriptional start sites as well as internal cleavage sites for known miRNA targets (S7E Fig). We found that almost all genes exhibiting increased mRNA abundance also exhibited increased 5′ ends at transcriptional start sites, but did not have reduced 5′ ends internal to the gene (S7D Fig). This was confirmed by visual observation of individual loci (S7E Fig). This suggests that the increase in the transcript abundance of these genes in *nrpd1−/−* was not caused by reduced mRNA cleavage compared to the WT. Only 5 genes—*PERK8*, *GLP2A*, *ETTIN*, *AAD3*, and *AT2G45245*—exhibited reduced cleavage at a few sites in *nrpd1−/−* endosperm. This minimal effect is contrary to the expectation that candidate genes described above should have reduced cleavage and suggests that sRNA-mediated posttranscriptional gene silencing is not a key mechanism for Pol IV to control endosperm gene expression. We therefore did not test if mRNA cleavage is impaired in *nrpd1+/−* heterozygotes.

## Assessing correspondence between sRNA and mRNA changes

Only a minority of the genes that have altered expression in *nrpd1* endosperm have associated changes in sRNAs within those same genes (S7A–S7C Fig). However, Pol IV sRNAs may also act at sites proximal to a gene to regulate it. We assessed the distance between misexpressed genes and altered sRNAs in homozygous and heterozygous *nrpd1* mutant endosperm. We found that 9.2%, 11.7%, and 3.3% of misregulated genes are within 1 kb of a site that loses sRNAs in *nrpd1−/−*, mat *nrpd1+/−*, and pat *nrpd1+/−* endosperm, respectively (S12 Table). To obtain a genome-wide perspective not focused on arbitrary distance cutoffs, we used the relative distance metric to test if genomic regions losing 24-nt sRNAs were associated with misregulated genes. The relative distance metric describes the spatial correlation between sRNA intervals and misregulated genes, compared to misregulated genes and random intervals [39]. This analysis found no enrichment in the association between Pol IV–dependent sRNAs and misregulated genes in any of the genotypes (S7F Fig).

## Assessing correspondence between DNA methylation and mRNA changes

The relevant molecular function of RNA Pol IV with regard to gene expression is typically assumed to be its role in RdDM. To identify potential examples of DNA methylation mediating Pol IV's impact on genes, we performed bisulfite sequencing of WT, mat *nrpd1+/−*, and pat *nrpd1+/−* endosperm DNA. We evaluated WT DNA methylation at individual cytosines within Pol IV sRNA-producing genes that were misregulated in *nrpd1−/−*, misregulated genes that showed increased sRNAs in *nrpd1−/−* (Pol IV-independent), and 5 control sets of genes that showed no change in sRNA abundance upon loss of Pol IV. We found that most cytosines were not methylated (median is near zero) in the genes we examined (S8A Fig). This suggests that Pol IV sRNAs do not generally target DNA methylation at genes. However, some misregulated genes with Pol IV–dependent sRNAs had higher CG methylation in WT endosperm (S8A Fig). These misregulated genes with Pol IV–dependent sRNAs were also likely to be longer (S8B Fig). Longer genes have higher sRNA read counts (S8C Fig), and, thus, differences in

these genes are more likely be called as statistically significant by DESeq2 [40]. Longer genes also tend to have higher CG methylation [41,42]. We therefore argue that this increased CG methylation is likely an analytical artifact. Our results suggest that Pol IV–dependent genic sRNAs do not regulate endosperm gene expression by directing genic DNA methylation, consistent with our previous findings [14].

We also tested if changes in DNA methylation brought about by loss of parental Pol IV could explain changes in gene expression. Overall, loss of parental Pol IV had only minor effects on DNA methylation (S9 Table). Loss of Pol IV activity primarily reduces asymmetric CHH methylation [43]. Comparison of mat *nrpd1*+/− and pat *nrpd1*+/− CHH methylation with WT endosperm identified 2,234 and 2,056 DMRs (covering 812.7 kb and 759.9 kb, respectively) with 50% hypomethylated in mat *nrpd1*+/− and 54.8% hypomethylated in pat *nrpd1*+/− (S9 Table). Consistent with the parent-of-origin effects described for mRNAs and sRNAs, we made 3 observations that suggest that mat and pat Pol IV activity have distinct impacts on the endosperm methylome. First, only 50% of CHH DMRs are shared between the 2 heterozygous genotypes. Second, regions where sRNA accumulation is dependent on paternal inheritance of a WT *NRPD1* allele have higher CHH methylation in WT endosperm than regions where sRNAs are dependent on maternal *NRPD1* (S8D Fig). This pattern is consistent with our previous finding that maternally biased sRNAs are often not associated with methylated DNA in WT endosperm [14]. Third, an examination of regions with at least 10% CHH methylation in WT endosperm shows that loss of paternal *NRPD1* had a more substantial impact on endosperm CHH methylation than loss of maternal *NRPD1* (S8E Fig).

Symmetric CG and CHG methylation are typically less affected by loss of *NRPD1* because other mechanisms exist to maintain this type of methylation. Comparison of CHG methylation between WT and either heterozygote identified fewer than 100 DMRs and was not investigated further. Both mat and pat *nrpd1*+/− endosperm exhibited changes in CG methylation compared to the WT (S9 Table). In mat *nrpd1*+/− endosperm, 48.5% of CG DMRs (of a total 600 DMRs spanning 207 kb) were hypomethylated relative to WT while in pat *nrpd1*+/−, 60% of CG DMRs (of a total 707 DMRs spanning 258 kb) were hypomethylated relative to WT. Further, we found that few of the sites hypo- or hypermethylated in the CG context in mat *nrpd1*+/− were shared with those changing methylation state in pat *nrpd1*+/− (S9 Table).

We used the DMRs in the analyses described above to assess their impact on gene expression. In mat *nrpd1*+/− endosperm, 2.6% and 3.4% of misregulated genes are within 1 kb of assayable regions with altered CHH methylation in mat *nrpd1*+/− endosperm (S12 Table). In addition, 2 genes and 1 gene are within 1 kb of a region that has higher and lower CHH methylation in pat *nrpd1*+/−. One gene is associated with increased CG methylation in pat *nrpd1*+/−. We also used relative distance analysis to see if mat *nrpd1*+/− misregulated genes are more likely to be associated with DNA methylation changes (there are too few genes associated with DNA methylation changes in pat *nrpd1*+/− to perform this analysis). Consistent with previous analyses, we find no clear relationship between DNA methylation changes and gene expression changes in the mat *nrpd1*+/− endosperm (S8F Fig).

## Allelic analysis of misregulated genes to identify *cis-* or *trans-*effects of Pol IV

One method to assess whether Pol IV's impacts on gene expression are predominantly *cis-* or *trans-*acting is to compare the allelic origins of mRNA in WT and *nrpd1*+/− endosperm. If a gene's mRNA abundance in the endosperm is determined by the activity of Pol IV in *cis* either in the gametophyte or sporophyte, then the gene would be primarily misregulated from the

allele inherited from a parent lacking Pol IV. Thus, in mat *nrpd1+/−* endosperm, misregulation of such genes would be driven predominantly by changes in expression from maternal alleles, whereas gene expression differences in pat *nrpd1+/−* endosperm would be driven by changes in expression from paternal alleles. In contrast, the predominance of *trans*-effects would be indicated by both parental alleles contributing to the changes in the total transcript abundance at most genes. We utilized SNPs between Col-0 (paternal) and L*er* (maternal) genomes to identify allele-specific mRNA-seq reads. We evaluated the contributions of each parent's alleles in the endosperm for 2372 misregulated genes that had at least 10 allele-specific reads in WT and mat *nrpd1+/−*. For the majority of genes, misregulation in mat *nrpd1+/−* was driven by effects on expression of both maternal and paternal alleles, with some notable exceptions (S9A Fig). For example, increased expression of *DOG1* in mat *nrpd1+/−* was primarily due to increased expression from maternal alleles (S9B Fig). The transcript from AT4G12870 was repressed in mat *nrpd1+/−* primarily due to a loss of maternal allele expression (S9B Fig). In contrast, expression of *SAC2* was primarily repressed in mat *nrpd1+/−* because of decreased expression from the paternal allele (S9B Fig). Overall, both maternal and paternal alleles made equal contributions to genic misregulation in the mat *nrpd1+/−* endosperm. Moreover, 4.7% of down-regulated genes and 5.3% of up-regulated genes showed at least a 20% increase or decrease in maternal allele contribution. This was roughly similar to the contribution of paternal alleles to misregulation in mat *nrpd1+/−*: 4.4% of down-regulated genes and 5.3% of up-regulated showed at least a 20% change in paternal allele contribution (S9A Fig). In pat *nrpd1+/−*, only 8% of down-regulated genes had lower contribution of paternal alleles, while both alleles contributed to up-regulation (S9A Fig). Overall, these results suggest that parental Pol IV's impact on gene expression is largely due to *trans*-effects.

In summary, our analyses test and dismiss several *cis*-regulatory mechanisms for how Pol IV may mediate parent-of-origin gene expression effects on the endosperm. We also individually examined DNA methylation and sRNAs at genes showing antagonistic regulation by maternally and paternally acting Pol IV and found no evidence for a role for sRNAs and DNA methylation in their regulation. These results lead us to conclude that the parent-of-origin and antagonistic effects that we observe are likely the result of *trans*-acting effects of parental Pol IV activity.

## Discussion

We demonstrate that Pol IV activity in the father promotes seed abortion in response to extra paternal genomes, whereas Pol IV activity in the mother promotes seed viability in these conditions. Previous observations of sRNA or mRNA from individual genes in diploid endosperm showed that Pol IV function in the mother and the father have different effects on the endosperm [11,44]. These findings suggested that Pol IV has differing, and perhaps even opposing, roles in maternal and paternal parents. In this study, we characterized the effect of maternal and paternal Pol IV activity on the endosperm through genome-wide analyses of transcription, mRNA cleavage, sRNAs, and DNA methylation in balanced endosperm. Our molecular data demonstrate that Pol IV activity in the mother and father have parent-of-origin effects on the endosperm, a subset of which are antagonistic. We found that one parent's copy of *NRPD1* is sufficient for the production of Pol IV–dependent sRNAs at most loci, with a small number of largely nonoverlapping loci losing sRNAs upon loss of maternal or paternal *NRPD1*. Pol IV activity in the mother and father also have distinct impacts on the endosperm DNA methylation landscape. Endosperm with a paternally inherited *nrpd1* mutation had lower DNA methylation compared with endosperm where the *nrpd1* mutation was maternally inherited. Finally, an interrogation of gene expression shows that loss of maternal Pol IV leads to

significant misregulation of several hundred genes, while loss of paternal Pol IV leads to misregulation of only several dozen. A key finding of our study is that genes that are misregulated upon loss of maternal *NRPD1* are affected in an opposite manner upon loss of paternal *NRPD1*. Together, our results suggest that maternal and paternal Pol IV are genetically antagonistic and that the major effect on transcription observed in heterozygotes is established before fertilization. These observations are important for understanding both Pol IV's role in reproduction and the genetic architecture underlying parental control of offspring development.

## Pol IV, conflict, and the genetic architecture of parental control

Parental conflict theory predicts that in viviparous and polyandrous species, mothers and fathers have antagonistic effects on regulating resource allocation and associated gene expression in offspring [45]. In practice, such effects are difficult to detect and have been infrequently described [46]. Analogous to observations for cryptic meiotic drive suppression systems [47], antagonistic parental effects are likely to be balanced in the individuals within an inbred population (like *Arabidopsis*) and are thus unobservable except in mutants or in hybrids where maternal and paternal effects are out of balance. When homozygous mutants are examined, these effects may be missed because they do not cause dramatic developmental phenotypes or because simultaneous loss of antagonistic maternal and paternal effects effectively cancels one another out. Thus, reciprocal heterozygotes need to be examined to detect antagonistic parent-of-origin effects. A close examination of our data provides insights into the genetic architecture mediating parental control of offspring development.

A key feature of the regulatory infrastructure that mediates parent-of-origin specific effects on zygotic gene expression is that maternal and paternal alleles need to be distinguished from each other in the zygote (in this case, endosperm is the relevant zygote). In *A. thaliana* endosperm, at many loci, maternally inherited alleles are DNA demethylated and marked with H3K27 methylation by polycomb repressive complex2 (PRC2), while paternally inherited alleles remain DNA methylated and have reduced H3K27me3 [21,48,49]. Maternal inheritance of mutations in the PRC2 subunits *MEA*, *FIE*, *FIS2*, and *MSI1* leads to endosperm defects and seed abortion [50–53]. Similarly, inheritance of maternal mutations in the DNA demethylase *DME* increases DNA methylation on endosperm maternal alleles and causes seed abortion [54]. Paternal inheritance of mutations in these genes have no reported effect on endosperm development or gene expression. These results thus argued that the solution to the problem of distinguishing parental alleles from one another after fertilization was to mark maternal and paternal chromosomes with distinct epigenetic modifications. However, this model may not explain all parent-of-origin effects on gene expression, particularly outside of imprinted genes. Our study provides evidence for a distinct model in which the same epigenetic regulator—Pol IV—can mediate both maternal and paternal effects. The only other example of a gene with seemingly antagonistic effects on seeds is the maintenance methyltransferase *MET1*, whose mutation has opposing effects on seed size when inherited maternally or paternally, although the molecular basis of this phenotype is unknown [55].

How does Pol IV in the mother and the father have distinct impacts after fertilization? Pol IV targets can be tissue or developmental stage specific [56], and, thus, Pol IV may target different genomic regions during male and female gametogenesis. Pol IV could act pre- or post-meiotically in the parental sporophyte (diploid phase of the life cycle), in the gametophyte (haploid phase of life cycle), or postfertilization in the maternal sporophyte. Reverse transcription PCR (RT-PCR)-based examination of dissected synergids and central cells did not detect *NRPD1* transcripts [44]. This suggests that on the maternal side, Pol IV influences endosperm gene expression by acting in the maternal sporophyte or in the female gametophyte prior to

central cell formation. Alternatively, Pol IV could act in the maternal sporophytic integuments/seed coat after fertilization, when the endosperm is developing. One potential mechanism for this would be through Pol IV–dependent sRNAs moving from the seed coat to the endosperm [11,56]. However, examination of the levels of total Pol IV–dependent sRNAs, allele-specific data, and ISRs suggests that the potential influence of seed coat Pol IV function on endosperm expression would likely be independent of sRNA transfer. This conclusion is consistent with previous observations that endosperm and seed coat have distinct sRNA profiles [14].

We have shown that parental Pol IV activity is dispensable for guiding endosperm sRNA production at most loci, with the exception of ISRs, but that parental Pol IV activity plays an important role in guiding endosperm gene expression. The molecular nature of this memory is unknown, and at present, we can only speculate. Data from paternal excess interploidy crosses suggest that the molecular identity of Pol IV memory may differ between the maternal and paternal parents. In the father, the genes required for sRNA production (*NRPD1*, *RDR2*, and *DCL3*) and the genes required for downstream DNA methylation (*NRPE1/Pol V* and *DRM2*) are both essential to promote paternal excess seed abortion [10]. In contrast, in the mother, genes required for sRNA production but not for DNA methylation promote paternal excess seed viability [10]. This suggests that DNA methylation or another downstream chromatin mark directed by Pol IV–dependent sRNAs could be the identity of paternally inherited memory, but is unlikely to be the molecular identity of maternally inherited memory. What would be the nature of maternal DNA methylation–independent memory? Pol IV, like other RNA polymerases [57], could act as a chromatin remodeler. Or, Pol IV could direct a chromatin modification, produce sRNAs that posttranscriptionally control genes, or control the expression of genes whose products are deposited in the gametes, which, in turn, sets up a memory to direct gene expression programs in the endosperm after fertilization.

How might we interpret Pol IV's parent-of-origin effects in terms of conflicts between parents? The WISO or "weak inbreeder/strong outbreeder" model [58] emerges from the dynamics of parental conflict and parent-of-origin effects. Under this model, a parent from populations with higher levels of outcrossing is exposed to higher levels of conflict and can thus dominate the programming of maternal resource allocation in a cross with an individual from a population with lower levels of outcrossing. Such a phenomenon has been observed in numerous clades including *Dalechampia*, *Arabidopsis*, *Capsella*, and *Leavenworthia* [59–62]. Intriguingly, loss-of-function phenotypes in the RdDM pathway are more severe in recently outcrossing species than in *A. thaliana* [12,13] and suggests that RNA Pol IV functions are more elaborate in these species. This raises the possibility that the role for RNA Pol IV and RdDM in parental conflict that we describe here in *A. thaliana* is likely heightened in and mediates the elevated level of parental conflict in species that are currently or have been recently outcrossing.

Studies on how resource allocation conflicts between parents impact gene expression have thus far been focused on imprinted genes. However, a handful of studies show the importance of nonimprinted genes in parent-of-origin effects [63,64]. For example, quantitative trait locus (QTL) analyses of a heterogeneous mouse stock showed that nonimprinted genes mediate parent-of-origin effects on the offspring's immune system [64]. Our study describes for the first time a system in which an epigenetic regulator acts in the mother and the father to antagonistically regulate the same nonimprinted genes in the zygote. While the magnitude of effects at many genes may be small, it should be noted that small changes in gene expression can be associated with very different phenotypes [65]. Our allele-specific mRNA-seq data show that loss of Pol IV from one parent can impact alleles inherited from both parents in the endosperm. This suggests that Pol IV does not act directly at antagonistic loci but acts instead by

regulating other modifiers of gene expression. Yet, this antagonistic regulation can also be viewed from another perspective. Parental conflict can be resolved or paused if both parents can modulate the expression level of a gene or the activity of a pathway to an optimum that is tolerable to each. Pol IV's role in mediating the antagonistic effects of both parents makes it an ideal system to negotiate optimal gene expression levels. Thus, Pol IV may not be solely an agent of conflict, but also a means to resolving it. Overall, these data suggest that Pol IV is part of a gene regulatory network that is evolving under parental conflict.

## Materials and methods

### *Arabidopsis* growth conditions, strains, and tissue collection

Plants used in this experiment were grown at 22°C in a Conviron chamber in 22-hour light (120 μM). The *A. thaliana* mutant used in this study was *nrpd1a-4* (SALK_083051 obtained from ABRC, The Ohio State University) [66] in the Col-0 background. We also utilized *nrpd1a-4* introgressed 4 times into L*er* [14]. Endosperm from approximately 100 seeds (7 days after pollination) from at least 3 siliques was dissected free of embryos and seed coats and pooled for each biological replicate as previously described [22]. Each biological replicate was collected from crosses that used different individuals as parents. The number of replicates for each experiment was decided based on currently accepted practices in genomic studies. For sRNA experiments, we planned to sample 3 biological replicates for each genotype. However, we had to discard one of the 3 pat *nrpd1+/−* sRNA libraries because that library had too few reads.

### mRNA, sRNA, and DNA isolation and library construction

Large- and small-sized RNAs were isolated using the RNAqueous-Micro RNA isolation kit (Thermo Fisher Scientific, Waltham, MA). Briefly, endosperm dissected from seeds was collected in lysis buffer and then homogenized with an RNAse-free pellet pestle driven by a Kimble motor. Large and sRNA species were isolated and separated using the manufacturer's protocol. The RNA concentration of the larger fraction was measured by Qubit. sRNA libraries were constructed using the NEXTFLEX Small RNA-seq kit v3 (Bioo Scientific, Austin, TX). Final library amplification was carried out for 25 cycles, and the libraries were size selected (135 to 160 bp) using a Pippin Prep (Sage Science, Beverly, MA). mRNA-seq libraries were constructed using a Smart-Seq2 protocol [67]. NanoPARE libraries were built as described in [38]. All libraries were sequenced on the Illumina Hi-Seq 2500. Seed coat contamination in our samples was ruled out by examining transcriptome data using a previously published tool [68].

DNA for bisulfite sequencing was isolated from dissected endosperm at 7 days after pollination using the QIAamp DNA Micro kit (QIAGEN 56304 Hilden, Germany). Dissected tissue was obtained for 2 biological replicates for each genotype and incubated overnight in a shaker at 56°C in ATL buffer with Proteinase K. Between 70 and 100 ng of endosperm DNA was subjected to bisulfite treatment using the MethylCode Bisulfite Conversion kit (Invitrogen, Waltham, MA). Analysis of cytosines from chloroplasts with at least 10 sequenced reads showed a conversion rate of greater than 98% for all libraries. Bisulfite converted DNA was used to build libraries with the Pico Methyl-Seq library kit (Zymo Research, D5455, Orange, CA). A total of 7 cycles of amplification were used for library construction. All libraries were sequenced on the Illumina Hi-Seq 2500 (60-bp paired-end reads).

### sRNA analysis

sRNA reads were trimmed with fastq_quality_trimmer (*fastq_quality_trimmer -v -t 20 -l 25*). Cutadapt [69] was used to identify adapter bearing reads of suitable length (*cutadapt -a*

*TGGAATTCTCGGGTGCCAAGG—trimmed-only—quality-base 64 -m 24 -M 40—max-n 0.5 —too-long-output*). Taking advantage of the random nucleotides on the adapters in NEXT-FLEX kits, we used Prinseq (prinseq-lite-0.20.4) (*prinseq-lite.pl -fastq <infile> -out_format 3 -out_good <filename> -derep 1 -log*) to remove PCR duplicates [70]. Filtered reads were aligned to a genome consisting of concatenated Col-0 TAIR10 and L*er* pseudo-genome (Col-0 genome substituted with L*er* SNPs) using Bowtie (v 1.2.2) *bowtie -v 2—best -p 8–5 4–3 4—sam <index file> <infile.fq>* (2 mismatches, report best alignment, ignore 4 bases on 5′ and 3′ ends) [71]. Reads mapping to L*er* were lifted over to Col-0 using custom scripts [14]. A custom script, assign-to-allele, was used to identify reads arising from Col-0 or L*er* alleles (https:// github.com/clp90/imprinting_analysis/tree/master/helper_scripts). Aligned reads between 21- and 24-nt in length were binned based on size. Bedtools was used to count reads in 300-bp windows with 20-bp overlaps and over annotated genes and TEs from Araport 11. DESeq2 [24] was used to identify features showing differences in sRNA abundance with an adjusted *p*-value of 0.05 or less. One complication with using DESeq2 is that the loss of Pol IV–dependent sRNAs at most loci in *nrpd1−/−* leads to an underestimation of WT library size by DESeq2, which increases the proportion of false negatives and undercounts the number of Pol IV–dependent sRNA loci. To allay this effect while analyzing genes, we excluded TEs and applied differential expression analysis to just genic and miRNA loci. These non-TE loci also included Pol IV-independent sRNA loci, which provide an estimate of library size. We separately examined TEs using genic sRNA counts to provide an estimate of library size. ShortStack version 3.8.5 [23] was also used as an orthogonal approach to identify sRNA peaks from bam alignment file output from Bowtie. Parameters chosen for ShortStack included dicermin = 20, dicermax = 25, and a mincov of 0.5 rpm. Weightage for multimapping reads was guided by uniquely mapping reads (option = u).

## mRNA-seq and NanoPARE analysis

The reads from mRNA-seq and NanoPARE were trimmed for quality with "*trim_galore -q 25 —phred64—fastqc—length 20—stringency 5*" and aligned to the TAIR10 genome using Tophat (v2.1.1) [72] using the command *tophat -i 30 -I 3000—segment-mismatches 1—segment-length 18 —b2-very-sensitive*. Cuffdiff (v2.1.1) [73] was used to identify differentially expressed genes for mRNA-seq data. Aligned NanoPARE read counts at each nucleotide in the genome were counted using bedtools. Sites with statistical differences in NanoPARE read counts were identified by DESeq2.

## DNA methylation analysis

Reads from Bisulfite sequencing were trimmed for quality using Trim Galore (https://github. com/FelixKrueger/TrimGalore). Trimmed reads were aligned to the TAIR10 genome using Bismark [74] with parameters set to *-N 1 -L 20—non_directional*. For this alignment, paired-end reads were treated as single reads. Previously described Bismark methylation extractor and custom scripts [21,75] were used to determine DNA methylation/base, and then methylation was calculated for 300-bp windows that overlapped by 200 bp. Data from the 2 biological replicates for each genotype were pooled together for comparison between genotypes. To be included in analysis, windows needed to have at least 3 common informational cytosines and a depth of 6 reads/cytosine. Windows that differed between genotypes by 10% CHH, 20% CHG, or 30% CG DNA methylation were identified as differentially methylated. Overlapping windows with differential methylation between genotypes were merged into differentially methylated regions. To increase the robustness of our conclusions, we added 2 data filtering steps. DNA methylation in the endosperm varies between maternal and paternal alleles and bisulfite

sequencing is known to potentially enrich for methylated DNA [76]. Since we were examining the consequences of loss of *NRPD1* in either parent, we could preferentially lose DNA methylation information from one set of alleles. This could lead to lower coverage of one set of parental alleles and lead to faulty measurements of DNA methylation. We therefore limited our analyses to genomic regions in which reads arising from the maternally inherited genome accounted for 67%+/− 15% of total DNA reads (based on the fact that 2/3 of the DNA in endosperm is maternally inherited). Next, we identified DMRs between the 2 replicates for each genotype to mark regions where DNA methylation was variable within the same genotype. These regions were excluded from further analysis.

## Supporting information

**S1 Fig. Maternal and paternal Pol IV activity have opposing effects on seed abortion caused by paternal genomic excess.** Loss of maternal *NRPD1* decreases paternal excess seed viability while loss of paternal *NRPD1* increases seed viability. Each dot in the aligned dot plot represents seed viability from one paternal excess cross (biological replicate). Significance of difference between indicated crosses was calculated by Wilcox test. Underlying data can be found in S1 Data. Pol IV, polymerase IV.
(PDF)

**S2 Fig. RNA Pol IV is necessary for the production of 21- to 24-nt sRNAs in the endosperm. (A)** Size (nt) of all sRNAs in endosperm sRNA peaks predominated by 21, 22, 23, or 24-nt sRNAs. ShortStack was used to call peaks in WT (L*er* × Col-0) endosperm. Each peak is grouped into a size class based on the predominant size of the sRNA species in that peak. Fraction of sRNAs at other sizes in the same peaks are plotted. **(B)** sRNA peaks of multiple sizes are impacted by loss of *NRPD1*. **(C)** Upset plot shows that genes losing sRNAs of one size class lose sRNAs of other size classes in *nrpd1*−/− endosperm. Data for S2A and S2B Fig can be found in S1 Data. Gene lists used for upset graph in S1C Fig can be extracted from GEO GSE197717. Pol IV, polymerase IV; sRNA, small RNA; WT, wild type.
(PDF)

**S3 Fig. Impact of the loss of maternal and paternal *NRPD1* on endosperm sRNA populations. (A)** One parent's copy of *NRPD1* is sufficient for 24-nt sRNA production from genes and TEs at most loci, here exemplified by *RIC5* and a *VANDAL21* insertion. **(B)** Examination of 21- to 24-nt sRNAs over genes and TEs shows that inheriting a maternal mutation in *NRPD1* has a greater impact than inheriting a paternal mutation in *NRPD1*. Loci with differential sRNA expression were identified using DESeq2. WT read counts represent average read counts per locus across 3 replicates. Reads mapping to TE insertions were normalized using genic sRNA expression. Black circles represent *padj* ≤ 0.05. Gray circles represent *padj* > 0.05. Data for this plot can be found in S2 Data. sRNA, small RNA; TE, transposable element; WT, wild type.
(PDF)

**S4 Fig. RNA Pol IV–dependent sRNAs arise from both maternal and paternal alleles.** SNPs between Col-0 and L*er* were used to identify the parental origins of sRNAs arising from genes and TEs. Differentially expressed loci were identified using DESeq2 as described in Fig 1. Loci with a sum of at least 10 allele-specific reads in 3 WT L*er* × Col-0 (WT) replicates and showing significant differences in 21-nt and 24-nt sRNAs in L*er nrpd1* −/− × Col *nrpd1*−/− endosperm were included. Box plots are Tukey plots. Numbers over box plots indicate the number of loci evaluated. Data represented in this figure can be found in S3 Data. Pol IV, polymerase IV;

sRNA, small RNA; WT, wild type.
(PDF)

**S5 Fig. Tissue enrichment in dissected endosperm shows little seed coat contamination.**
For each mRNA-seq library built with RNA from dissected endosperm, reads overlapping
genic loci were counted with Htseq-count. Enrichment of a seed tissue in each sample was
then calculated using the tissue enrichment tool [68]. mRNA-seq, mRNA sequencing.
(PDF)

**S6 Fig. Impact of Pol IV on imprinted gene expression and imprinting. (A)** A subset of
imprinted genes are misregulated by loss of maternal or all *NRPD1*. Loss of paternal *NRPD1*
has limited impact on expression. Scatter plots show output from Cuffdiff calculating the dif-
ference in gene expression between WT and indicated mutant genotype. Black circles repre-
sent genes whose abundance varies by 2-fold and q < 0.05. All other genes represented by gray
circles. **(B)** Aligned dot plot representing fold change for imprinted genes exhibiting signifi-
cant differences in expression. **(C)** Allele-specific expression is not impacted at most imprinted
loci. % maternal of all Col-L*er* imprinted genes identified in [21] was calculated by counting
reads overlapping Col/L*er* SNPs. **(D)** Examples of imprinted genes whose allelic bias was
impacted by loss of all *NRPD1*. In WT, *WOX8* and *SAC2* are predominantly expressed from
maternal and paternal alleles, respectively. In *nrpd1−/−*, *WOX8* is down-regulated because of
reduced expression from the maternal allele, while the expression of *SAC2* is driven by down-
regulation of the paternal allele. Data represented in this figure can be found in S5 Data. Pol
IV, polymerase IV; WT, wild type.
(PDF)

**S7 Fig. Little relationship between Pol IV sRNAs and gene regulation. (A–C)** Comparisons
of genes showing significant differences in 21-, 22-, 24-nt sRNA and mRNA abundance shows
that only a subset of genes (lower right quadrant) may be repressed by Pol IV–dependent
sRNAs in WT. Differences in sRNA abundance between WT and *nrpd1−/−* were calculated
using DESeq2. Differences in mRNA was calculated using Cuffdiff. Numbers in bold in each
quadrant indicate number of genes. **(D)** NanoPARE data maps 5′ ends of transcripts and iden-
tifies TSSs and cleavage sites within the gene body. Change in mRNA cleavage at genes that
show increased mRNA abundance and decreased 21-, 22-, or 24-nt sRNAs. Coverage of 5′ end
reads from NanoPARE sequencing was calculated for every nucleotide in the genome. Differ-
ence in 5′ read coverage at each nucleotide was calculated for 2 replicates of WT endosperm
(L*er* × Col) and 3 *nrpd1−/−* (L*er nrpd1−/−* × Col *nrpd1−/−*) replicates using DESeq2. Each
point plotted on the dot plot represents one nucleotide with differential 5′ reads overlapping a
gene. A single gene may thus have more than one 5′ read mapping region. **(E)** Examination of
NanoPARE data from 2 replicates of WT and *nrpd1−/−* correctly identifies a documented
*miR159* cleavage site in *MYB65* but identifies no difference in putative cleavage of the
*YUCCA10* transcript. *YUCCA10* was chosen as an example because it shows increased mRNA
abundance and reduced sRNA abundance in *nrpd1−/−*. **(F)** The relative distance metric shows
no significant correlation between misregulated genes and sites losing sRNAs in *nrpd1+/−* and
*nrpd1−/−*. Relative distance was calculated using bedtools. Black line indicates relative distance
between sites losing sRNA (identified by DESeq2 by examination of read counts over 300-bp
windows) and misregulated genes. Gray lines represent 5 replicates of equivalent number of
random sites in the genome and misregulated genes. A uniform frequency of about 0.02 indi-
cates no major correlation between the 2 datasets. 5,896, 1,720, and 790 sites lost sRNAs in
*nrpd1−/−*, mat *nrpd1+/−*, and *pat nrpd1+/−*, respectively. The relative choppiness of the distri-
bution in pat *nrpd1+/−* is likely driven by the smaller number of sites being compared. Data

represented in this figure can be seen in S6 Data. Pol IV, polymerase IV; sRNA, small RNA; TSS, transcriptional start site; WT, wild type.
(PDF)

**S8 Fig. No relationship between DNA methylation changes and genic misregulation in *nrpd1*+/−. (A)** Comparison of methylation in CG, CHG, and CHH contexts at individual cytosines within genes in WT. Cytosines in the first 2 columns on the left lie within misregulated genes whose sRNA abundances are up or down in *nrpd1*−/−. The control sets include cytosines within 5 randomly selected subset of genes that show no changes in mRNA abundance in *nrpd1*−/−. **(B)** Genes with fewer sRNAs in *nrpd1*−/− and misregulated expression tend to be longer. **(C)** Longer genes have more total 24-nt sRNAs in WT endosperm. **(D)** WT CHH methylation at sRNA-producing sites that are dependent on maternal or paternal Pol IV. Methylation is significantly higher at paternal Pol IV–dependent sites. **(E)** Effects of parental Pol IV loss on CHH methylation at regions with parental Pol IV–dependent sRNAs and greater than 10% CHH methylation in WT. Red, difference between mat *nrpd1*+/− and WT; blue, difference between pat *nrpd1*+/− and WT. sRNA-producing regions impacted in paternal *nrpd1*+/− have greater losses of CHH methylation. For D and E, CHH methylation was calculated for 300-bp sliding windows with a 200-bp overlap. CHH methylation windows overlapping windows losing sRNAs in *nrpd1*+/− endosperm were identified and merged using bedtools; maximum CHH methylation among merged windows was used for violin plot. *** represents a statistically significant difference as calculated by Wilcoxon test ($p < 0.001$). Boxplot in the violin plot shows median and interquartile range. **(F)** The relative distance metric shows no significant correlation between misregulated genes and sites with changes in CG and CHH DNA methylation in mat *nrpd1*+/−. Relative distance was calculated using bedtools. Black line indicates relative distance between misregulated genes and sites with differences in DNA methylation between WT and mat *nrpd1*+/− (identified by Bismark). Gray lines represent relative distance between 5 replicates of random sites in the genome and misregulated genes. A uniform frequency of about 0.02 indicates no major correlation between the 2 datasets. Data represented in this figure can be found in S7 Data. Pol IV, polymerase IV; sRNA, small RNA; WT, wild type.
(PDF)

**S9 Fig. Impact of parental *NRPD1* on maternal and paternal allele contributions to total gene expression. (A)** Genes were examined to identify those whose expression differences were driven by allele-specific effects. Genes with at least a 2-fold, statistically significant difference in expression between the indicated heterozygote and WT and at least 10 allele-specific reads in both genotypes were included. The shift in allelic expression was evaluated by subtracting the % maternal allele transcripts in WT from the heterozygote. Genes within Col-0 introgressions that remain in Ler *nrpd1*−/− plants were excluded from all analyses. **(B)** Examples of genes showing allele-specific impacts upon loss of maternal Pol IV. FPKM and fold change in (A) and (B) are from Cuffdiff output. Data represented here can be found in S8 Data. FPKM, fragments per kilobase of exon per million; Pol IV, polymerase IV; WT, wild type.
(PDF)

**S1 Table. List of sequenced libraries.**
(XLSX)

**S2 Table. DESeq2 output for comparison of 21- and 24-nt sRNAs over genes between WT and *nrpd1*−/−.** sRNA, small RNA; WT, wild type.
(XLSB)

**S3 Table. DESeq2 output for comparison of 21- and 24-nt sRNAs over TEs between WT and *nrpd1*−/−.** sRNA, small RNA; TE, transposable element; WT, wild type.
(XLSB)

**S4 Table. DESeq2 output for comparison of 21- and 24-nt sRNAs over genes between WT and maternal *nrpd1*+/−.** sRNA, small RNA; WT, wild type.
(XLSB)

**S5 Table. DESeq2 output for comparison of 21- and 24-nt sRNAs over TEs between WT and maternal *nrpd1*+/−.** sRNA, small RNA; TE, transposable element; WT, wild type.
(XLSB)

**S6 Table. DESeq2 output for comparison of 21- and 24-nt sRNAs over genes between WT and paternal *nrpd1*+/−.** sRNA, small RNA; WT, wild type.
(XLSB)

**S7 Table. DESeq2 output for comparison of 21- and 24-nt sRNAs over TEs between WT and paternal *nrpd1*+/−.** sRNA, small RNA; TE, transposable element; WT, wild type.
(XLSB)

**S8 Table. Bedgraph for windows with differences in 21- and 24-nt sRNAs between WT and *nrpd1*−/−, maternal *nrpd1*+/−, and paternal *nrpd1*+/−.** sRNA, small RNA; WT, wild type.
(XLSB)

**S9 Table. Bed files showing regions differentially methylated between WT, mat *nrpd1*+/−, and pat *nrpd1*+/− endosperm.** WT, wild type.
(XLSB)

**S10 Table. Cuffdiff output showing genes that are differentially expressed between WT, *nrpd1*−/−, mat *nrpd1*+/−, and pat *nrpd1*+/−.** WT, wild type.
(XLSB)

**S11 Table. Gene ontology analysis for genes misregulated in mat and pat *nrpd1*+/−.**
(XLSB)

**S12 Table. Distance between misregulated genes and regions with differences in sRNAs and DNA methylation.** sRNA, small RNA.
(XLSB)

**S1 Data. Data underlying Fig 1 (analysis of sRNA populations in *nrpd1*−/−, mat *nrpd1*+/−, and pat *nrpd1*+/−), S1 Fig (seed abortion levels in interploid crosses), and S2 Fig (analysis of Shortstack data).** sRNA, small RNA.
(XLSX)

**S2 Data. Data underlying S3 Fig.** Comparisons of 21- to 24-nt sRNAs over genes and TEs in WT with *nrpd1*−/−, mat *nrpd1*+/−, and pat *nrpd1*+/−. sRNA, small RNA; TE, transposable element; WT, wild type.
(XLSB)

**S3 Data. Data underlying Fig 2 and S4 Fig.** Analysis of the allelic origins of sRNAs in *nrpd1*−/−, mat *nrpd1*+/−, and pat *nrpd1*+/−. sRNA, small RNA.
(XLSX)

**S4 Data. Data underlying Fig 3.** Analysis of gene expression changes in *nrpd1*−/−, mat *nrpd1*+/−, and pat *nrpd1*+/−.
(XLSB)

**S5 Data. Data underlying S6 Fig.** Analysis of imprinted gene expression in *nrpd1*−/−, mat *nrpd1*+/−, and pat *nrpd1*+/−
(XLSX)

**S6 Data. Data underlying S7 Fig.** Analysis of the relationship between changes in sRNA and mRNA abundance. sRNA, small RNA.
(XLSX)

**S7 Data. Data underlying S8 Fig.** Analysis of the relationship between changes in DNA methylation and mRNA abundance.
(XLSX)

**S8 Data. Data underlying S9 Fig.** Analysis of the impacts of mat and pat *nrpd1*+/− on allelic contributions to mRNA abundance.
(XLSX)

# Acknowledgments

We thank the Whitehead Institute Genome Technology Core for high-throughput sequencing services and Dr. Michael Nodine for advice on NanoPARE. This research was supported by National Science Foundation Awards 2101337 and 1453459 to M.G.

# Author Contributions

**Conceptualization:** Prasad R. V. Satyaki, Mary Gehring.

**Data curation:** Prasad R. V. Satyaki.

**Formal analysis:** Prasad R. V. Satyaki, Mary Gehring.

**Funding acquisition:** Mary Gehring.

**Investigation:** Prasad R. V. Satyaki, Mary Gehring.

**Methodology:** Prasad R. V. Satyaki, Mary Gehring.

**Supervision:** Mary Gehring.

**Visualization:** Prasad R. V. Satyaki, Mary Gehring.

**Writing – original draft:** Prasad R. V. Satyaki, Mary Gehring.

**Writing – review & editing:** Prasad R. V. Satyaki, Mary Gehring.

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
