## [Editor Report · Decision Letter 0]

6 Jan 2022

Dear Dr Gehring, 

Thank you for submitting your revised manuscript entitled "RNA Pol IV has antagonistic parent-of-origin effects on Arabidopsis endosperm" for consideration as a Research Article by PLOS Biology along with the Review Commons reports. Thank you also for your patience as we completed our editorial process, and please accept my apologies again for the delay in providing you with our decision due to the Christmas holidays.

Your manuscript has now been evaluated by the PLOS Biology editorial staff as well as by an academic editor with relevant expertise and I am writing to let you know that we would like to send your submission back to the original reviewers. However, we would like to consider it as a Short Report. Please select this article type from the dropdown menu when you complete the metadata.

Before we can send your manuscript to the reviewers, we need you to complete your submission by providing the metadata that is required for full assessment. To this end, please login to Editorial Manager where you will find the paper in the 'Submissions Needing Revisions' folder on your homepage. Please click 'Revise Submission' from the Action Links and complete all additional questions in the submission questionnaire.

Once your full submission is complete, your paper will undergo a series of checks in preparation for peer review. Once your manuscript has passed the checks it will be sent out for review. To provide the metadata for your submission, please Login to Editorial Manager (https://www.editorialmanager.com/pbiology) within two working days, i.e. by Jan 10 2022 11:59PM.

If your manuscript has been previously reviewed at another journal, PLOS Biology is willing to work with those reviews in order to avoid re-starting the process. Submission of the previous reviews is entirely optional and our ability to use them effectively will depend on the willingness of the previous journal to confirm the content of the reports and share the reviewer identities. Please note that we reserve the right to invite additional reviewers if we consider that additional/independent reviewers are needed, although we aim to avoid this as far as possible. In our experience, working with previous reviews does save time. 

If you would like to send previous reviewer reports to us, please email me at ialvarez-garcia@plos.org to let me know, including the name of the previous journal and the manuscript ID the study was given, as well as attaching a point-by-point response to reviewers that details how you have or plan to address the reviewers' concerns. 

Given the disruptions resulting from the ongoing COVID-19 pandemic, please expect some delays in the editorial process. We apologise in advance for any inconvenience caused and will do our best to minimize impact as far as possible.

Kind regards,

Ines

--

Ines Alvarez-Garcia, PhD

Senior Editor

PLOS Biology

---

## [Decision Letter · Decision Letter 1]

9 Feb 2022

Dear Dr Gehring,

Thank you for submitting your revised Short Report entitled "RNA Pol IV has antagonistic parent-of-origin effects on Arabidopsis endosperm" for publication in PLOS Biology. I have now obtained advice from the original Review Commons reviewers and have discussed their comments with the Academic Editor. 

Based on the reviews (attached below), we will probably accept this manuscript for publication, provided you satisfactorily address the remaining points raised by Reviewer 2. Please also make sure to address the following data and other policy-related requests.

In addition, we would like you to consider a suggestion to improve the title:

"RNA Pol IV induces antagonistic parent-of-origin effects on Arabidopsis endosperm"

We expect to receive your revised manuscript within two weeks. 

*Published Peer Review History*

*Early Version*

Sincerely,

Ines

--

Ines Alvarez-Garcia, PhD,

Senior Editor,

PLOS Biology

Fig. 1A, B, D-F; Fig. 2A-F; Fig. 3B-H; Fig. S1; Fig. S2A-C; Fig. S3B; Fig. S4; Fig. S6A-D; Fig. S7A-D, F; Fig. S8A-F and Fig. S9A, B

In addition, please provide the accession numbers of the sequencing data submitted to NCBI GEO and make them publicly available at this stay.

BLURB

Please also provide a blurb which (if accepted) will be included in our weekly and monthly Electronic Table of Contents, sent out to readers of PLOS Biology, and may be used to promote your article in social media. The blurb should be about 30-40 words long and is subject to editorial changes. It should, without exaggeration, entice people to read your manuscript. It should not be redundant with the title and should not contain acronyms or abbreviations. For examples, view our author guidelines: https://journals.plos.org/plosbiology/s/revising-your-manuscript#loc-blurb

Reviewers' comments

Rev. 1:

I could quibble about how functional significant some of these differences are, but the authors have essentially satisfied my concerns.

Rev. 2: Thomas Städler – note that this reviewer has signed the review

This manuscript is a revised version of a manuscript that I previously reviewed for Review Commons. In my mind, it was a solid piece of work back then, and now has been revised based on the extensive comments by the three original referees. Overall, the Results section has been successfully streamlined, and the Discussion section has been expanded to better reflect the implications of this work. There are, however, some remaining issues that I would like to have addressed before publication.

(1) I am curious as to the omission of parts of all reviewers' comments from the "Response to Referees" part of this submission. For example, in my case, only comments that I listed under the sections "Major Comments" and "Minor Comments" have been replicated and addressed by the authors, whereas the required sections "Evidence, reproducibility and clarity" and "Significance" have been left out. Exactly the same applies to the feedback given by referee 3, and some of the minor comments made by referee 2 are "missing". I can understand that my longish and generally very positive assessment previously made in my "Summary" has gone without specific responses, but I did make specific suggestions in the "Significance" part that I would like to have addressed. I could understand why the authors might feel that including my suggestions would make their paper too speculative, but this should be explicitly stated in the "Responses".

(2) Although I specifically requested that all cited references should include volume and page (or article) numbers, as is the rule for all journals that I am familiar with, Satyaki & Gehring have not delivered, despite their assurances. I count 12 references with such problems, and 2 of those are listed with a single author when in reality there are 4 authors (Grossniklaus et al. 1998) and 6 authors (Köhler et al. 2003), respectively. In addition, in most cases genus and species names are not italicized in the References section, plus a few other formatting issues.

Minor Comments

* Page 3, lines 17-21: the explanation of the rationale underlying the kinship model does not appear to be entirely correct. There might be cases where the inclusive fitness of mothers is not optimal under equal provisioning (e.g. if fathers differ in genetic quality and mothers can somehow sense this); the model's rationale is that relatedness of mothers to all of her offspring is identical, while under a polygamous mating system fathers sire offspring that compete for finite maternal resources with their half-sibs.

* Page 4, 1st paragraph: at first occurrence in the main text, the names of genera should be spelled out (Arabidopsis, Brassica, Capsella).

* Page 16, 1st paragraph of section "Pol IV, conflict, ….": the beginning of this section needs a few citations, as well as more details. I don't think that all viviparous species would qualify; the relationship asymmetry obtains under multiple paternity or, more generally, scales with the degree of relatedness between (multiple) parents.

* Page 18, line 7: technically, Brandvain & Haig (2005) called it "weak inbreeder/strong outbreeder" model.

* Page 18, lines 6-19: this constitutes the new paragraph that the authors inserted in response to my previous comments. There is nothing "wrong" with it but I think it is omitting some recent developments suggesting that the WISO hypothesis needs to be broadened to include differences in levels of parental conflict among obligate outcrossers. In other words, it now appears that differences in the breeding system are not the only factor influencing levels of parental conflict, as recent empirical data in Mimulus (Coughlan et al. 2020, Curr. Biol.) and Solanum (Roth et al. 2019, Genetics; Städler et al. 2021, Curr. Opin. Plant Biol.) suggest that larger differences in effective population size between hybridizing, outbreeding lineages result in greater asymmetry in seed development and levels of seed failure. These aspects are at the core of what I suggested to consider in my previous referee comments (alluded to above under major point 1).

* Page 20, line 9: substitute "were" for "was" at end of line.

* Page 22, line 4: there is a duplicate "that" before "2/3".

* References: the above-mentioned incomplete information concerns Blevins et al. 2015, Gehring et al. 2011, Ji et al. 2014, Kim et al. 2013, Martin 2011, Mi & Thomas 2009, Milbocker & Sink 1969, Panda et al. 2020, Pignatta et al. 2014, 2015.

* Figure S5: the figure legend mentions Schon & Nodine 2017, but the full citation is not given anywhere.

Rev. 3: Steven E. Jacobsen – note that this reviewer has signed the review

The authors have made all of the recommended changes and the paper is ready for publication.

---

## [Editor Report · Decision Letter 2]

11 Mar 2022

Dear Dr Gehring,

On behalf of my colleagues and the Academic Editor, Xuemei Chen, I am pleased to say that we can in principle accept your Short Reports "RNA Pol IV induces antagonistic parent-of-origin effects on Arabidopsis endosperm" for publication in PLOS Biology, provided you address any remaining formatting and reporting issues. These will be detailed in an email that will follow this letter and that you will usually receive within 2-3 business days, during which time no action is required from you. Please note that we will not be able to formally accept your manuscript and schedule it for publication until you have any requested changes.

PRESS

Sincerely, 

Ines

--

Ines Alvarez-Garcia, PhD 

Senior Editor 

PLOS Biology
